# Facile and Rapid Synthesis of Porous Hydrated V_2_O_5_ Nanoflakes for High-Performance Zinc Ion Battery Applications

**DOI:** 10.3390/nano12142400

**Published:** 2022-07-14

**Authors:** Kai Guo, Wenchong Cheng, Haoxiong Chen, Hanbin Li, Jinxue Chen, Haiyuan Liu, Yunliang Tu, Wenhao She, Zhengkai Huang, Yinpeng Wan, Lixia Zou, Zhuyao Li, Xing Zhong, Yongchuan Wu, Xianfu Wang, Neng Yu

**Affiliations:** 1Jiangxi Province Engineering Research Center of New Energy Technology and Equipment, School of Chemistry, Biology and Materials Science, East China University of Technology, Nanchang 330013, China; wenchongcheng@ecut.edu.cn (W.C.); chenhx@ecut.edu.cn (H.C.); hanbinli@ecut.edu.cn (H.L.); chenjinxue@ecut.edu.cn (J.C.); haiyuanliu@ecut.edu.cn (H.L.); yunliangtu@ecut.edu.cn (Y.T.); liming9889@ecut.edu.cn (W.S.); huangzhk@ecut.edu.cn (Z.H.); 2020330464@ecut.edu.cn (Y.W.); lxzou@ecut.edu.cn (L.Z.); lizhuyao@ecut.edu.cn (Z.L.); zhongxing@ecut.edu.cn (X.Z.); 2State Key Laboratory of Materials Processing and Die & Mould Technology, Huazhong University of Science and Technology, Wuhan 430000, China; 3Jiangxi Key Laboratory for Mass Spectrometry and Instrumentation, East China University of Technology, Nanchang 330013, China; wuyoungchan@163.com; 4State Key Laboratory of Electronic Thin Film and Integrated Devices, University of Electronic Science and Technology of China, Chengdu 610054, China

**Keywords:** hydrated vanadium pentoxide, nanoflakes, zinc ion battery, facile synthesis, room temperature

## Abstract

Hydrated V_2_O_5_ with unique physical and chemical characteristics has been widely used in various function devices, including solar cells, catalysts, electrochromic windows, supercapacitors, and batteries. Recently, it has attracted extensive attention because of the enormous potential for the high-performance aqueous zinc ion battery cathode. Although great progress has been made in developing applications of hydrated V_2_O_5_, little research focuses on improving current synthesis methods, which have disadvantages of massive energy consumption, tedious reaction time, and/or low efficiency. Herein, an improved synthesis method is developed for hydrated V_2_O_5_ nanoflakes according to the phenomenon that the reactions between V_2_O_5_ and peroxide can be dramatically accelerated with low-temperature heating. Porous hydrated V_2_O_5_ nanoflake gel was obtained from cheap raw materials at 40 °C in 30 min. It shows a high specific capacity, of 346.6 mAh/g, at 0.1 A/g; retains 55.2% of that at 20 A/g; and retains a specific capacity of 221.0 mAh/g after 1800 charging/discharging cycles at 1 A/g as an aqueous zinc ion battery cathode material. This work provides a highly facile and rapid synthesis method for hydrated V_2_O_5_, which may favor its applications in energy storage and other functional devices.

## 1. Introduction

The increasing market demand for various commercial electronics, electric vehicles, and large-scale energy storage has triggered an urgent need for rechargeable batteries with low cost, high energy density, high power density, and good safety [1,2,3,4,5]. Lithium-ion batteries have become the most popular energy storage devices for the past two decades [6,7,8]. Nevertheless, alternative aqueous rechargeable batteries are attracting more and more interest due to their low potential production cost, earth-abundant elements, and inflammable electrolytes. Among them, zinc ion batteries (ZIBs) demonstrate several distinctive merits, such as the theoretical specific capacity of 819 mAh/g; the low redox potential, of −0.76 V, (vs. standard hydrogen electrode); nontoxicity; and great stability in water [9,10]. However, ZIBs still face several challenges, such as low energy density, poor rate performance, and fast capacity decay. High-performance cathode materials, reliable zinc metal anodes, and efficient electrolytes are required.

Tremendous efforts have been devoted to exploiting high-performance cathode materials for aqueous ZIBs. Several cathode materials have been successfully synthesized, such as MnO_2_, Prussian blue analogues, Na_3_V_2_P(O4)_3_, and vanadium-based compounds [11,12,13,14,15,16,17,18]. Among them, V_2_O_5_·nH_2_O (hydrated V_2_O_5_) is one of the most promising cathode materials due to unique intrinsic properties, including multiple stable valences; a bilayer structure; adjustable interlayer space, between 8.8 and 17.2 Å (corresponding *n* between 0.5 and 6); and one of the highest theoretical capacities, of 510 mAh/g (when *n* = 1.6), for discovered ZIB cathode materials [19,20]. The substantial interlayer space and crystalline water endow it with excellent zinc ion storage capability [21,22,23]. For example, Yan et al. synthesized a V_2_O_5_·nH_2_O/graphene composite material for a ZIB cathode, which displayed a specific capacity of 372 mAh/g at 0.3 A/g [22]. Mg-ion-doped V_2_O_5_·nH_2_O developed by Ming and co-workers delivers high capacities, of 353 and 264 mAh/g, at current densities of 100 and 1000 mA/g, respectively, along with long-term durability [24]. In addition, the hydrated V_2_O_5_ is an important material for supercapacitors, other ion storage systems, solar cells, catalysts, and electrochromic windows [25,26,27,28,29]. Although great progress has been achieved in exploiting applications in various functional devices, hardly any effort has been devoted to developing highly efficient and green synthesis methods. 

The most common synthesis method is the solution method, including hydrolysis and condensation from alkoxide, ionic exchange, quenching of molten V_2_O_5_, electrodeposition, and the reaction of V_2_O_5_ with peroxide [30]. Hydrolysis and condensation from alkoxide precursors, such as VO(OC_3_H_7_)_3_, are generally preferred due to good control of reacting species concentration but require expensive raw materials and long reaction time (such as 4 days) [31]. Quenching of molten V_2_O_5_ can synthesize hydrated V_2_O_5_ quickly and involves melting V_2_O_5_ powder at 700–800 °C and then quenching in water [28]. The ionic exchange method uses an ion exchange resin for the acidification of metavanadate solution, leading to additional water consumption and un-exchanged Na^+^ or NH_4_^+^ ion [32,33,34,35]. Hydrated V_2_O_5_ on the mg/cm^2^ scale can also be electrodeposited on a conductive substrate in VOSO_4_ solution, followed by annealing at 120 °C for 20 h [29]. The reaction of V_2_O_5_ with peroxide is a common method because of cheap raw materials and high-purity products. However, this method requires a high-temperature hydrothermal reaction for 8–10 h or an ambient environment reaction for several days for complete gelation [23,36]. Anyway, the above synthesis methods are successful but have some disadvantages, such as expensive raw materials, high energy consumption, tedious reaction time, and/or low efficiency. Green chemistry has already aroused strong interest because it is important for sustainable development [37,38,39]. Therefore, it is valuable to explore facile and efficient synthesis methods for hydrated V_2_O_5_. 

This work presents an advanced route for the hydrated V_2_O_5_ nanomaterial. High-quality hydrated V_2_O_5_ was obtained in just 30 min by gently elevating the environment temperature of the reaction between V_2_O_5_ and peroxide to 40 °C. Although this is a minor modification to the traditional method, reaction duration or temperature is greatly reduced. The obtained hydrated V_2_O_5_ has a porous nanoflake morphology with a large surface area. It was mixed with carbon nanotubes (CNTs) to construct a layer-structured composite film and used as an aqueous zinc ion battery cathode directly. The hydrated V_2_O_5_ shows a high specific capacity, of 346.6 mAh/g, at 0.1 A/g and still retains 55.2% of that when the discharging current density is increased to 20 A/g; and its capacity remains 221.0 mAh/g after 1800 charging/discharging cycles at 1 A/g. Electrochemical kinetics analysis suggests that pseudocapacitance plays a dominant role in the zinc ion storage of hydrated V_2_O_5_. 

## 2. Experimental Section

### 2.1. Synthesis of Hydrated V_2_O_5_


In all, 0.72 g of V_2_O_5_ powder (Sinopharm Chemical, China) was dissolved in the mixture solution of 4 mL hydrogen peroxide solution (10%, Sinopharm Chemical, China) and 60 mL deionized H_2_O under sonication for 15 min at 40 °C to form a clear reddish solution. The dissolution process was exothermic, while hydrogen peroxide was partially decomposed and released oxygen gas in the meantime. The reddish solution was then heated to 80 °C by a hot plate under stirring. After 20 min, the clear solution slowly became a dark-red viscous gel. After drying in the air under 80 °C for 8 h, a dark-red dry gel was obtained. The obtained dry gel was then mixed with a CNT solution (10 mg/mL, XFNANO, Nanjing, China) and then filtered to form a freestanding composite film, in which the mass ratio of hydrated V_2_O_5_ was 70%. The CNT in the composite film serves as electric conducting networks, similar to the carbon material in the conventional powder electrode.

### 2.2. Materials Characterization

The as-prepared hydrated V_2_O_5_ dry gel and composite film were characterized by X-ray powder diffraction (XRD on a Bruker D8 Advance X-ray diffractometer with Cu Kα radiation; λ = 1.54060 Å; scan range 5–80°). The morphologies and structures were examined by using a field-emission scanning electron microscope (SEM, Nova NanoSEM 450, FEI, Lincoln, NE, USA) equipped with an energy-dispersive X-ray (EDX) detector. The elemental states in the dry gel were revealed with an X-ray photoelectron spectrometer (XPS, PHI QUANTERA-II SXM, Ulvac-PHI, Chigasaki, Japan). Transmission electron microscopy (TEM) analysis was conducted on a Titan G260-300 instrument. Thermogravimetric analysis (TGA, Discovery TGA 5500, New Castle, DE, USA) of the sample was carried out at a ramping rate of 10 °C/min from room temperature to 700 °C in the air atmosphere.

### 2.3. Electrochemical Measurements

All electrochemical tests were conducted on a CR2025-type coin cell at room temperature. The mass loading of active material per electrode is about 1 mg/cm^2^. All cells were assembled in air with a V_2_O_5_/CNT composite film cathode, a zinc foil (100-μm-thick) anode, and a glass fiber membrane separator, and 3 M Zn(CF_3_SO_3_)_2_ (Adamas, China) was used as the electrolyte. Cyclic voltammetry (CV) tests were conducted on a CHI 660E electrochemical workstation. The galvanostatic charge/discharge (GCD) test and the galvanostatic intermittent titration technique (GITT) test were carried out on a NEWARE 4000 testing system at room temperature. For the GITT test, the coin cell was charging/discharging for 20 min at 0.1 A/g with 120 min relax duration. The solid diffusion coefficient was calculated according to the equation below.
(1)$$D\;=\frac{4L^{2}}{\pi \tau }{\left(\frac{∆E_{s}}{∆E_{t}}\right)}^{2}$$
where t, τ, and $∆E_{s}$ represent the duration of the current pulse (s), the relaxation time (s), and the steady-state voltage change (V) induced by the current pulse, respectively [40]. $∆E_{t}$ is the voltage change (V) during the galvanostatic current pulse after eliminating the IR drop. L is the ion diffusion length (cm) of the electrode, which equals to the thickness of the hydrated V_2_O_5_/CNT composite film electrode (13.9 μm). All the specific capacities of the V_2_O_5_ samples are calculated based on the mass of V_2_O_5_ in the composite films.

## 3. Results and Discussion

The preparation process of hydrated V_2_O_5_ nanoflakes is illustrated in Figure 1. Firstly, V_2_O_5_ powder was added to the peroxide solution and dissolved under stirring at 40 °C for 15 min, leading to a transparent reddish solution. During the dissolution, a large amount of heat was released, accompanied by the generation of gas bubbles. The gas is regarded as oxygen derived from the decomposition of vanadium peroxo species according to previous studies [36,41,42]. The reddish solution then slowly changed to a viscous dark-red hydrated V_2_O_5_ gel after another 15 min of stirring at 40 °C. The phenomenon was derived from a series of complicated chemical reactions between V_2_O_5_ and H_2_O_2_, including the formation and successive decomposition of vanadium peroxo species, according to previous literature [36,41,43]. After the gel was dried at 80 °C for 8 h, a dark-red dry V_2_O_5_ gel was obtained. In contrast, the V_2_O_5_ powder can be dissolved in the H_2_O_2_ solution under stirring at room temperature in about 30 min and the obtained reddish gel remains nearly unchanged in the following 24 h at room temperature. Anyway, it seems heating can accelerate the gelation process of hydrated V_2_O_5_. As far as we know, this phenomenon has not been reported yet. The method proposed in this work improves the room temperature gelation method (involving the reaction between V_2_O_5_ and H_2_O_2_) by extensively shortening the synthesis time of several days to less than 30 min through the consumption of ignorable energy [36]. This improved method has some advantages compared to previous methods. For a better comparison, the typical synthesis conditions of the reported hydrated V_2_O_5_ materials are collected and listed in Table 1. Previous methods have the disadvantages of expensive raw materials, long reaction time, high reaction temperature, and/or low efficiency, while this method has obvious advantages of low cost, high efficiency, high production, and environmental friendliness at the same time. It makes this improved method an attractive and high-potential method for the large-scale synthesis of hydrated V_2_O_5_.

The crystalline structure and composition of hydrated V_2_O_5_ were characterized by XRD and XPS. The XRD pattern of the as-prepared sample is shown in Figure 2a. All the peaks are similar to the (00n) planes of hydrated V_2_O_5_ (JCPDS NO. 40-1296). The interlayer of the (001) plane (d_001_ = 12.3 Å) indicates that the number of crystalline water molecules in the hydrated V_2_O_5_·nH_2_O is close to 1.9 according to the Van der Waals diameter of the water molecule (2.7 Å) [19]. TGA was conducted to determine the ratio of water molecules in hydrated V_2_O_5_, as shown in Appendix A. Similar to literature data, the TGA plot shows three weight loss stages, corresponding to three types of bound water molecules [19,23]. From 30 to 120 °C, the weight loss is generally attributed to weakly bound water molecules. The weight loss between 120 and 280 °C comes from departure of more strongly bound water molecules, and complete water molecules are removed when hydrated V_2_O_5_ is heated to 400 °C, along with a total weight loss of 15.4%, corresponding to 1.8 moles of H_2_O per mole of V_2_O_5_·nH_2_O [19,23]. The XPS spectra (Figure 2b) present four peaks, in which the two major ones, located at the binding energies of 517.6 eV (V 2p_3/2_) and 525.0 eV (V 2p_1/2_), correspond to the spin-orbit peaks of the V–O bond from V^5+^ in the hydrated V_2_O_5_ and the other two, minor ones, are attributed to the spin-orbit peaks of V–O bond from V^4+^ [23,28]. The XPS spectra indicate that the dominant state valence of the V element in the hydrated V_2_O_5_ is +5, while the other few V elements are in the state of V^4+^. The presence of V^4+^ is common in hydrated V_2_O_5_ [28,44,45]. The specific surface area of the V_2_O_5_ sample is 14.7 m^2^/g, and its pore size distribution is presented in Figure 2c. The pores in V_2_O_5_ have a diameter in the range of 1–18 nm, and the majority is in the range of 1–8 nm, indicating there are substantial micro- and mesopores in the hydrated V_2_O_5_ sample. The morphology of hydrated V_2_O_5_ was characterized by SEM and TEM. Figure 2d shows that the hydrated V_2_O_5_ has a morphology of nanoflakes. The TEM image suggests that the V_2_O_5_ nanoflakes are composed of stacked thin nanosheets (Figure 2e). The selected area electron diffraction pattern in the inset shows the polycrystalline nature of the V_2_O_5_ nanoflakes. The high-resolution TEM image in Figure 2f demonstrates that there are two nanopores in the nanosheet, confirming the nanoporous morphology. 

The hydrated V_2_O_5_ nanoflakes were dispersed into CNT aqueous solution and then vacuum-filtered to obtain a piece of composite film, as shown in Appendix A. The cross-section SEM image displays that the composite film has a layered structure with a thickness of 13.9 μm (Appendix A). The composite film is denser than the pure CNT film due to the addition of hydrated V_2_O_5_ nanoflakes, and it seems hydrated V_2_O_5_ nanoflakes are mixed evenly with CNT, as shown in Appendix A. The distribution of hydrated V_2_O_5_ nanoflakes is characterized by EDX. Appendix A displays that the carbon element from CNT and oxygen and vanadium elements from hydrated V_2_O_5_ are uniform in the selected area, confirming an even distribution of hydrated V_2_O_5_ in the composite film.

The electrochemical performance of hydrated V_2_O_5_ as the cathode of an aqueous zinc ion battery is evaluated by coin-type cells, using the hydrated V_2_O_5_/CNT composite film cathode, a Zn foil anode, and 3 M Zn(CF_3_SO_3_)_2_ aqueous electrolyte. The CV curves of V_2_O_5_ between 0.2 and 1.5 V versus Zn/Zn^2+^ at a scan rate of 0.1 mV/s demonstrate multistep reactions and contain three pairs of redox peaks, located at 0.74/0.59, 1.03/0.89, and 1.14/0.94 V (Figure 3a). This electrochemical behavior is similar to that of V_2_O_5_·nH_2_O in previous work [21]. The charging branches of all the three CV plots have three oxidation peaks in the same potential ranges, and the CV plots of the second and third cycles nearly overlap, indicating the electrochemical reactions in V_2_O_5_ are reversible. The charging and discharging plots of the initial 5 GCD curves at 0.1 A/g have the same reaction plateaus in identical potential ranges, confirming that the electrochemical reactions are reversible (Figure 3b). The electrochemical performance of the hydrated vanadium oxide has been thoroughly studied as a cathode of a zinc ion battery, and research results indicate highly reversible Zn^2+^ ion intercalation/de-intercalation during the discharging/charging process, which can be expressed by the following equation [22,23,46,47]:(2)$$V_{2}O_{5}\cdot nH_{2}O\;+xZn^{2+}\to \;Zn_{x}V_{2}O_{5}\cdot nH_{2}O$$

Figure 3c demonstrates the superior rate capability of the hydrated V_2_O_5_ sample. It has specific capacities of 346.6, 312.5, 301.6, 292.5, 236.9, 208.3, and 191.2 mAh/g at current densities of 0.1, 0.5, 1, 5, 10, 15, and 20 A/g, respectively. Moreover, the specific capacity retention is 55.2% when the current density increases from 0.1 to 20 A/g. The specific capacity and rate performance are comparable or superior to recent hydrated-V_2_O_5_-based cathode materials in other literature, such as the V_2_O_5_·nH_2_O nanoflake/graphene composite (372 mAh/g and 248 mAh/g at 0.3 and 30 A/g, respectively) [22], Mg_x_V_2_O_5_·nH_2_O nanobelts (353 and 183 mAh/g at 0.3 and 1 A/g, respectively) [24], Zn_0.25_V_2_O_5_·nH_2_O nanobelts (282 and 260 mAh/g at 0.1 and 6 A/g, respectively) [17], and Ca_0.24_V_2_O_5_·0.83H_2_O nanobelts (289 and 72 mAh/g at 0.55 and 44A/g, respectively) [48]. The faint plateaus in the charging/discharging plots at 20 A/g in Figure 3c indicate the existence of pseudocapacitive capacity, and its proportion is appreciable at high-speed charging/discharging. The stable discharge capacity platforms even at 20 A/g (Figure 3d) suggest a stable structure of the hydrated V_2_O_5_ material, which is verified by the cycling test. Figure 3e demonstrates that the hydrated V_2_O_5_ displays an initial discharge capacity of 361.2 mAh/g and retains a discharge capacity of 221.0 mAh/g after 1800 charging/discharging cycles at 1 A/g, indicating satisfactory long-term cycling performance. Meanwhile, the columbic efficiency remains stable at 100% in the whole cycling test, further confirming highly reversible reactions.

The electrochemical kinetics of hydrated V_2_O_5_ was investigated by measuring its CV curves at various scan rates and conducting a GITT test. Figure 4a shows that the CV curves maintain similar shapes, while the oxidation and reduction peaks shift to higher and lower voltages at increasing scan rates between 0.1 and 1.5 mV/s, respectively. According to the calculation equation in the literature, the peak current–scan rate relationship was fitted [40,49]. The calculated *b* values for peaks 1 to 4 are 0.99, 0.89, 0.91, and 0.88, respectively (Figure 4b). The *b* values are far away from 0.5 and close to 1, indicating the dominant proportions of pseudocapacitive contribution in the hydrated V_2_O_5_ sample. It is confirmed by the calculated capacitive-controlled contribution of 78.3, 85.3, 89.3, 91.8, 93.5, and 95.4% at scan rates of 0.1, 0.2, 0.5, 0.8, 1, and 1.5 mV/s, respectively (Figure 4c). High pseudocapacitive contribution in the capacity can be attributed to its unique porous stacked nanosheet morphology with abundant surface-active sites. The Zn^2+^ solid-state diffusion in the lattice of hydrated V_2_O_5_ was analyzed by the GITT test. Figure 4d shows the typical charging/discharging GITT curves after two ordinary cycles at a current density of 0.1 A/g. Each GITT curve contains 10 charging/discharging–relaxation processes for the full voltage window between 0.2 and 1.5 V. The galvanostatic charging/discharging process lasts 20 min, and then the cell relaxes for 120 min to allow the voltage to come to an equilibrium, as shown in Figure 4e. The Zn^2+^ diffusion coefficients (D_Zn_) in the hydrated V_2_O_5_ derived from the GITT data in Figure 4d are between 4 × 10^−12^ and 3 × 10^−11^ (Figure 4f). The high diffusion coefficient can be attributed to the layer structure with interlayer crystalline water and a large interlayer space of 12.3 Å, which are believed to be beneficial to the deintercalation of zinc ions [15,21,22]. 

## 4. Conclusions

In summary, porous hydrated V_2_O_5_ nanoflake gel was successfully synthesized through a facile stirring mixture of V_2_O_5_ powder and peroxide solution at 40 °C in just 30 min. The as-prepared hydrated V_2_O_5_ as a cathode active material for the zinc ion battery exhibited a maximum specific capacity of 346.6 mAh/g at 0.1 A/g, 55.2% capacity retention from 0.1 to 20 A/g, and excellent cycling performance. The performance is comparable or superior to the results of previous hydrated-V_2_O_5_-based materials in other literature. Electrochemical kinetics study suggests that pseudocapacitance plays a dominant role in the zinc ion storage process. The good performance of hydrated V_2_O_5_ can be attributed to its porous nanoflake morphology, with abundant active sites and high zinc ion diffusion coefficient in the materials. It is concluded that this near-room temperature synthesis method is a facile, rapid, and cost-effective route for highly qualified hydrated V_2_O_5_ and may promote its research and applications in energy storage and other functional devices.

## Figures and Tables

**Figure 1 nanomaterials-12-02400-f001:**
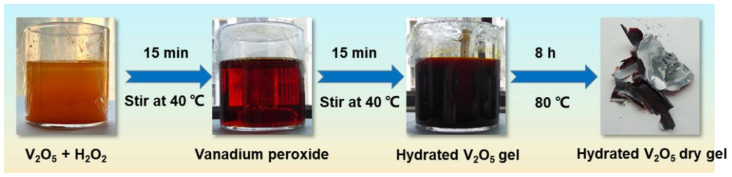
Schematic synthesis process of the V_2_O_5_ dry gel.

**Figure 2 nanomaterials-12-02400-f002:**
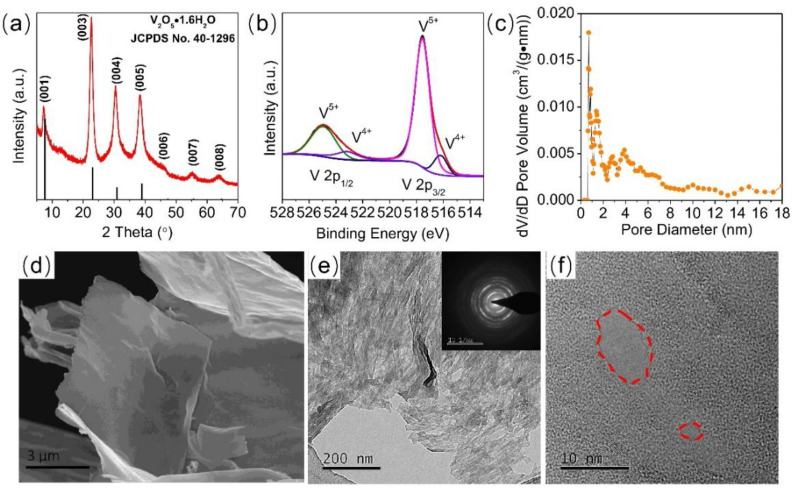
The structural analysis, chemical compositions, and micro-structure characterizations of hydrated V_2_O_5_. (**a**) XRD analysis, (**b**) XPS spectra, (**c**) pore size distribution, (**d**) SEM, (**e**) TEM, and (**f**) high-resolution TEM. Inset: selected area electron diffraction pattern.

**Figure 3 nanomaterials-12-02400-f003:**
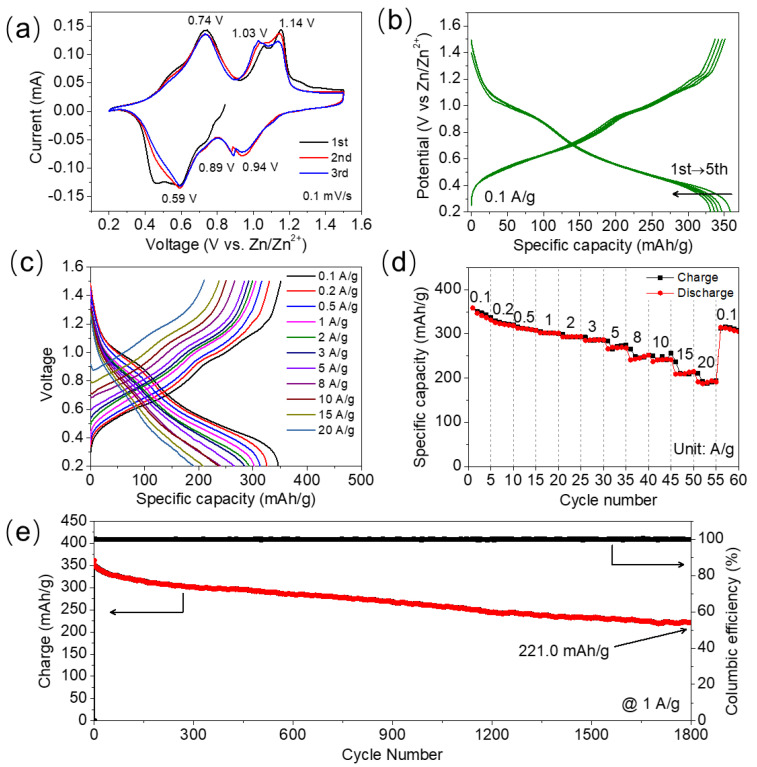
The electrochemical performance of hydrated V_2_O_5_. (**a**) The initial three CV curves at 0.1 mV/s, (**b**) the initial five GCD curves at 0.1 A/g, (**c**) the charging/discharging plots at current densities of 0.1 to 20 A/g, (**d**) the charging/discharging specific capacities at current densities of 0.1 to 20 A/g, and (**e**) the cycling test results for 1800 times at 1 A/g.

**Figure 4 nanomaterials-12-02400-f004:**
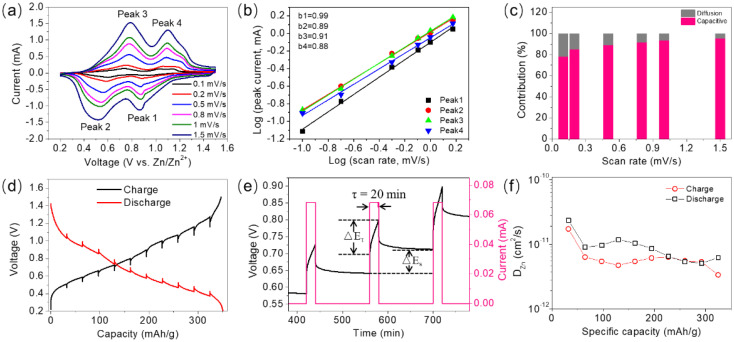
Electrochemical kinetics study of hydrated V_2_O_5_. (**a**) CV curves at various scan rates between 0.1 and 1.5 mV/s and (**b**) fitted relationship of peak currents with scan rates. (**c**) Calculated pseudocapacitance and diffusion-controlled capacities at different scan rates. (**d**) GITT plots after two ordinary charging/discharging cycles at 0.1 A/g. (**e**) Selected steps of the GITT curve during charging. (**f**) The corresponding Zn^2+^ diffusion coefficients (D_Zn_) during charging and discharging.

**Table 1 nanomaterials-12-02400-t001:** Representative routes for the synthesis of hydrated V_2_O_5_ materials in previous literature.

Vanadium Sources	Other Reagents	Temperature	Time	Ref.
VO(OC_3_H_7_)_3_	H_2_O, acetone	RT *	4 days	[31]
V_2_O_5_	H_2_O	800 °C	1–2 h	[28]
NaVO_3_	Resin, H_2_O	RT *	3 days	[35]
V_2_O_5_	H_2_O_2_, H_2_O	205 °C	14 h	[23]
V_2_O_5_	H_2_O_2_, H_2_O	RT *	26 h	[36]
VOSO_4_	H_2_O	120 °C	20 h	[29]
V_2_O_5_	H_2_O_2_, H_2_O	40 °C	0.5 h	This work

*: room temperature.

## Data Availability

Not applicable.

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
