# Peer review of "Facile and Rapid Synthesis of Porous Hydrated V2O5 Nanoflakes for High-Performance Zinc Ion Battery Applications"

_nanomaterials, 2022, doi:10.3390/nano12142400_

Round 1
Reviewer 1 Report
In this manuscript, the authors designed a facile method for the synthesis of porous hydrated V2O5 nanoflakes and demonstrated their outstanding performance as cathode material for Zn-ion battery (ZIB). The physical and chemical properties of the materials are demonstrated by various characterizations including XRD, XPS, SEM, TEM, etc. When evaluated as cathode materials in ZIB, the material presents superb rate performance at high current density. I would like to recommend the acceptance of this manuscript after minor revision. Here are some questions.
1. The application of hydrated oxides for ZIB cathodes is quite interesting. Will the hydrated V2O5 be partially dehydrated during the cycling? Otherwise, will the anhydrous V2O5 be hydrated during the cycling?
2. The temperature window of the new Zn-ion battery system should be indicated. How much will the performance be affected at sub-zero temperature or above 50 degrees?
3. How strong is the interaction between V2O5 and CNT?
4. The authors have employed ultrahigh current density of 20 A/g during the rate performance test. The sample present a superb specific capacity of 200 mAh/g in this condition. Why is the material so stable under such high current density? The thickness of the electrode might also be indicated.
Reviewer 2 Report
Guo et al present a new route to porous and hydrated V2O5 via the simple elevation of the peroxide solution temperature to 40oC. The aper is well organised and the material is well characterised. I recommend this paper for publication after the following changes have been made.
· Why was 40oC chosen as the temperature?
· Why 15 mins?
· Please provide reasoning for why the elevated temperature is helping. Is it doing more than just allowing the reaction to complete more quickly?
· There authors need to compare the results of their route (material characterisation & functional testing) to the results achieved via the some of the methods in the literature. For example how does the hydrated V2O5 produced from this study compare from the hydrated V2O5 produced in ref 23, 28 & 36? Please compare and discuss the XRD, XPS morphology and electrochemical performance.
· Figure 2b. The x axis in the XPS graph needs to go from high BE to low BE as it is in eV.
